# Antimicrobial activity of silver-copper coating against aerosols containing surrogate respiratory viruses and bacteria

Lorena Reyes-Carmona[1,2], Omar A. Sepúlveda-Robles[3], Argelia Almaguer-Flores[1]*, Juan Manuel Bello-Lopez[4], Carlos Ramos-Vilchis[5], Sandra E. Rodil[5]*

1 Laboratorio de Biointerfases, DEPeI, Facultad de Odontología, Universidad Nacional Autónoma de México, CDMX, México, 2 Programa de Maestría y Doctorado en Ciencias Médicas Odontológicas y de la Salud, Facultad de Odontología, Universidad Nacional Autónoma de México, CDMX, México, 3 Unidad de Investigación Médica en Genética Humana, UMAE Hospital de Pediatría, Centro Médico Nacional "Siglo XXI", Instituto Mexicano del Seguro Social (IMSS), CDMX, México, 4 Dirección de Investigación, Hospital Juárez de México, Magdalena de las Salinas, CDMX, México, 5 Instituto de Investigaciones en Materiales, Universidad Nacional Autónoma de México, CDMX, México

* srodil@unam.mx (SER); aalmaguer@comunidad.unam.mx (AAF)

**Data Availability Statement:** All relevant data are within the paper and its Supporting Information files.

## Abstract

The transmission of bacteria and respiratory viruses through expelled saliva microdroplets and aerosols is a significant concern for healthcare workers, further highlighted during the SARS-CoV-2 pandemic. To address this issue, the development of nanomaterials with anti-microbial properties for use as nanolayers in respiratory protection equipment, such as face-masks or respirators, has emerged as a potential solution. In this study, a silver and copper nanolayer called SakCu® was deposited on one side of a spun-bond polypropylene fabric using the magnetron sputtering technique. The antibacterial and antiviral activity of the AgCu nanolayer was evaluated against droplets falling on the material and aerosols passing through it. The effectiveness of the nanolayer was assessed by measuring viral loads of the enveloped virus SARS-CoV-2 and viability assays using respiratory surrogate viruses, including PaMx54, PaMx60, PaMx61 (ssRNA, *Leviviridae*), and PhiX174 (ssDNA, *Microviridae*) as representatives of non-enveloped viruses. Colony forming unit (CFU) determination was employed to evaluate the survival of aerobic and anaerobic bacteria. The results demonstrated a nearly exponential reduction in SARS-CoV-2 viral load, achieving complete viral load reduction after 24 hours of contact incubation with the AgCu nanolayer. Viability assays with the surrogate viruses showed a significant reduction in viral replication between 2–4 hours after contact. The simulated viral filtration system demonstrated inhibition of viral replication ranging from 39% to 64%. The viability assays with PhiX174 exhibited a 2-log reduction in viral replication after 24 hours of contact and a 16.31% inhibition in viral filtration assays. Bacterial growth inhibition varied depending on the species, with reductions ranging from 70% to 92% for aerobic bacteria and over 90% for anaerobic strains. In conclusion, the AgCu nanolayer displayed high bactericidal and antiviral activity in contact and aerosol conditions. Therefore, it holds the potential for incorporation into personal protective equipment to effectively reduce and prevent the transmission of aerosol-borne pathogenic bacteria and respiratory viruses.

**Funding:** This study was financially supported by Secretaría de Educación, Ciencia, Tecnología e Innovación (SECTEI) [www.sectei.cdmx.gob.mx] in the form of a grant (096/2020) received by SER. This study was also financially supported by Dirección General de Asuntos del Personal Académico (DGAPA) of the Universidad Nacional Autónoma de México (UNAM) [www.dgapa.unam.mx] in the form of a project (Programa de Apoyo a Proyectos de Investigación e Innovación Tecnológica (PAPIIT)) award (IT201121) received by AA-F. This study was also financially supported by Consejo Nacional de Humanidades, Ciencia y Tecnología (CONAHCYT) [www.conahcyt.mx] in the form of a PhD scholarship award (CVU 917708) received by LR-C. The funders had no role in study design, data collection and analysis, decision to publish, or preparation of the manuscript.

**Competing interests:** The authors have declared that no competing interests exist.

## Introduction

Several pathogenic bacteria and viruses could be transmitted by aerosols formed by saliva droplets expelled when people talk, cough, or sneeze. After the COVID-19 pandemic, society is aware of the need to carry out regular hand and surface disinfection processes, and during infection outbreaks or indoor activities [1], the use of facemasks will be required. The bio-aerosols are the main source of transmission of respiratory microorganisms, including the severe acute respiratory syndrome coronavirus-2 (SARS-CoV-2) [2–5], coronaviruses (SARS and MERS), rubeola virus (measles) [6], influenza virus [7, 8] or varicella-zoster virus (chickenpox) [9] which can be transmitted in hospitals [10, 11] or even in the environment [12]. Moreover, healthcare professionals are continuously exposed to pathogenic microorganisms during routine and surgical procedures [13–17]. In this regard, the use of personal protection equipment (PPE) such as coats, facemasks, respirators, and others, are essential kits to prevent several infectious diseases transmitted by aerosols [18, 19]; however, the PPEs by themselves do not reduce the viability of pathogenic microorganisms.

In recent years, significant progress has been made in the field of antimicrobial surfaces, focusing on utilizing nanomaterials and nanotechnology to combat the spread of infectious respiratory and oral diseases, such as COVID-19 [20–22]. A short summary of the antimicrobial materials to combat SARS-CoV-2 is shown in **Table 1**.

It can be observed that metals such as Silver (Ag), copper (Cu) and their bimetallic combination (AgCu) have been proved to be one of the most effective materials used to reduce the viability of different bacteria and the inactivation of several viruses [24, 39]. Ag has several antibacterial and antiviral mechanisms of action, such as membrane degradation, nucleic acid damage, disruption of proteins, oxidative stress, and others [40, 41]. While Cu also presents an antimicrobial mechanism denominated "contact killing" in which bacteria and viruses are killed in a short time [42–44].

In a previous study, we reported the effectiveness of a nanometric AgCu film (called SakCu®) deposited on both sides of a 0.3 mm (± 0.03 mm) thick polypropylene [45] fabric, which is usually used as a filtration material in the PPEs, including the N95 masks. The results showed that this nanolayer has virucidal and bactericidal properties against the SARS-CoV-2 virus and pathogenic bacteria associated with pneumonia (ESKAPE). AgCu film was not cytotoxic to human fibroblasts and keratinocytes [46]. However, the results were related only to the effect of the nanolayer on drops containing the virus or bacteria. Considering the importance of the microorganisms travelling in aerosols, in this work, we have developed a methodology to evaluate the antimicrobial properties of the AgCu nanolayer against virus and bacteria loaded aerosols. Due to safety concerns, the evaluation of the virus containing aerosols was done using surrogate models of respiratory non-enveloped viruses PaMx54, PaMx60, PaMx61 (ssRNA, *Leviviridae*) and PhiX174 (ssDNA, *Microviridae*). These bacteriophages are considered suitable viral surrogates for studying viruses that infect eukaryotic cells because they present similar structural characteristics and are safe to use. Also, they are relatively easy to produce in large quantities and suitable for antiviral studies [47–50].

Additionally, the antibacterial capacity of the AgCu nanolayer (SakCu®) to inhibit the growth of aerosols containing bacteria was tested using aerobic bacteria (*Escherichia coli*, *Pseudomonas aeruginosa*, *Staphylococcus aureus*, *Staphylococcus epidermidis*), and oral anaerobic

**Table 1. A short summary of materials proposed and tested for bacteria and respiratory viruses such as SARS-CoV-2.**

| Material | Viral or bacterial pathogen | Antimicrobial activity | Ref. |
|---|---|---|---|
| Ag-NPs PVD-coated Ag-NPs | TGEV-Coronavirus SARS-CoV-2 | Antiviral properties on TGEV infection Highly potent microbicides | [23, 24] |
| | Monkeypox | Antiviral effect | [25] |
| | RSV-Virus | 44% Inhibition of viral activity | [26] |
| | Gram-positive | Antibacterial activity | [27] |
| | Gram-negative | | |
| | bacteria | | |
| Cu surfaces Cu-Ag coating | Human Norovirus | Virus degradation in less than 5 min | [28] |
| | | Virus loses its capsid integration | [29] |
| | SARS-CoV-2 | Virus inhibition at $\leq$ 4 h | [30] |
| | *Staphylococcus aureus* | Filtration efficiency attributed by the composition of the Cu-Ag (65–35%) | [31] |
| CuS coating | SARS-CoV-2 | The virus is highly inactivated within 30 min exposure | [32] |
| IO-NPs | SARS-CoV-2 | Hinders virus adsorption | [33] |
| ZnO-NPs | SARS-CoV-2 | Decrease in viral load (70–90%) | [34, 35] |
| | | Induce severe viral damage | |
| | Nosocomial respiratory and oral bacteria | Higher bacteria inhibition or elimination | [36, 37] |
| TiO$_2$ Nanomaterials | SARS-CoV-2 | Antibacterial and antiviral potential due to its photocatalytic properties | [38] |
| | Gram-positive | | |
| | Gram-negative | | |
| | bacteria | | |

Nanoparticles (NPs), Transmissible gastroenteritis virus (TGEV), Polyvinylpyrrolidone (PVD), respiratory syncytial virus (RSV), Copper sulfide (CuS), Iron Oxide (IO), Zinc oxide (ZnO), Titanium dioxide (TiO$_2$).

bacteria (*Porphyromonas gingivalis, Aggregatibacter actinomycetemcomitans serotype b, Streptococcus mutans, Actinomyces israelii*).

Since a minor modification was done to the production of the AgCu nanolayer in comparison to the previous work [46], we did also test the viral inactivation and bacteria inhibition of respiratory droplets, called the On-contact method, where droplets containing bacteria or viruses were placed on the surface of the coated and uncoated PP and allowed to dry, mimicking the contamination of the PPE surfaces by respiratory droplets. The study also included the evaluation of the inactivation of SARS-CoV-2 virus and ESKAPE bacteria using the On-contact method.

## Materials and methods

### AgCu nanolayer deposition and characterization

A bimetallic 50 at.% Ag - 50 at.% Cu target (4", 99.99% purity) (Plasmaterials) was used for the deposition of the AgCu nanolayer using magnetron sputtering [46]. The deposition chamber was a homemade roll-to-roll system, where a 70 g m$^{-2}$ PP (Montblan corporation) roll passed (4" diameter and 18 cm width) in front of the sputtering target at 6 cm distance and a velocity of 6 rpm. The PP roll was placed inside the vacuum chamber to achieve $8 \times 10^{-6}$ Torr; then, Ar was introduced at a flow rate of 8 sccm, and the pressure was adjusted to 26 mTorr. The plasma was initiated using a radio frequency (RF) source at 200 W to maintain a low deposition rate. Circles of 1 cm diameter uncoated PP (control) and AgCu-coated PP named SakCu® (experimental) were cut and used for all subsequent tests [46]. There are two different aspects with respect to our previous work [46]. One is that a single target (50:50 Ag:Cu) was used instead of a Cu target with silver wires. The second important difference is the

deposition on a single face of the polypropylene instead of deposition on both faces used in the previous work. As shown here, deposition of both faces is not really necessary to obtain the antimicrobial properties.

## Filtration efficiency (NaCl)

The filtration efficiency of the uncoated and coated PP was tested using NaCl particles of different sizes (0.3, 0.5, and 1 μm). Larger particles are easily trapped, so we concentrate on the small size droplets. Genomic fragments of the SARS-CoV-2 virus can be found in any particle size, but Liu et al. showed a larger fraction in 0.25–0.5 μm droplets [51]. Additionally, considering that smaller droplets can travel distances greater than the recommended 1–2 meters for social distancing actions, the 0.3–1.0 μm range was considered appropriate. A homemade system was used to estimate the filtration efficiency, consisting of a NaCl particle generator, two optical particle counters (OPC), a vacuum chamber, and a mechanical pump, all placed inside a hood (not in operation). The design and operation were based on the proposal from Drewnick et al [52]. One OPC counts the particles in the hood, while the other counts the particles that pass through the filter material when the vacuum pump is activated for 6:30 minutes at a face velocity of 1.4 m/s. Filtration efficiency is obtained as:

$$FE = 100 \times (1 - \frac{p_f}{p_{air}})$$

(1)

where, $P_f$ is the average number of particles of a given diameter that pass through the material, and $P_{air}$ is the corresponding number inside the hood. Five measurements of each textile (20 s) were performed, and the experiments were repeated using three pieces (65 mm in diameter) for each material.

## Antimicrobial tests

**Surrogate viruses.** To demonstrate the antiviral properties of the AgCu nanolayer, bacteriophages (surrogate viruses) were used as model viruses due to their safety and experimental efficiency [47, 48, 50]. Four bacteriophage strains were used as surrogate models of respiratory viruses (**Table 2**). Sepúlveda-Robles and cols previously isolated PaMx54, PaMx60, PaMx61 bacteriophages [53]. While the phage PhiX174 was purchased from the ATCC collection (13706-B1).

Propagation of phages was performed by confluent lysis using the double-layer plaque assay technique for enumeration of virus surrogates [54] (**Fig 1**). Briefly, ~$10^5$ phages and 300 μL of an overnight host bacteria culture (*Pseudomonas aeruginosa* Ps33 for PaMx54, 60 and 61, and *E. coli rfaB* mutant for PhiX174) were mixed in a sterile test tube, incubated for 10 min at room temperature to allow adsorption of the phage to host. Then, 3 mL of soft Tφ medium (10g Bacto-tryptone, 5g NaCl, 7g Bacto-agar, 2M MgCl$_2$) was added and poured into a LB plaque to form a bacterial lawn. This was incubated at 37˚C overnight for confluent lysis of the bacteria. Finally, the top agar layer was scraped off and incubated overnight at 4˚C in 5 mL of phage buffer (50 mM Trisma-base pH 8, 10 mM MgSO$_4$, 100 mM NaCl, 2% gelatin), centrifuged at 10,509 g for 10 min, and the supernatant or phage lysate was recovered [53]. Finally, each phage lysate was titled by serial dilution on a bacterial host lawn and quantified as a "Plaque-Forming Unit (PFUs)", which is a measure of the quantity of viruses (bacteriophages) that are capable of infecting and lysing host cells as shown in **Fig 1**.

**ATCC bacteria species.** Four aerobic and four anaerobic bacterial strains from the American Type Cell Culture Collection (ATCC) were used for the antibacterial test and are listed in **Table 3**.

**Table 2. Surrogate viruses used in this study.**

| Bacteriophage | PaMx54 | PaMx60 | PaMx61 | PhiX174 |
|---|---|---|---|---|
| Host | *P. aeruginosa* | *P. aeruginosa* | *P. aeruginosa* | *E. coli* |
| Genome | ssRNA | ssRNA | ssRNA | ssDNA |
| Family | *Leviviridae* | *Leviviridae* | *Leviviridae* | *Microviridae* |
| Capsid | Non-enveloped, Icosahedral | Non-enveloped, Icosahedral | Non-enveloped, Icosahedral | Non-enveloped, Icosahedral |
| Size (nm) | 25 | 25 | 25 | 25 |
| References | [53] | | | ATCC 13706-B1 |

The aerobic strains were individually cultured on agar plates with Trypticase Soy Agar (TSA) (BBL, Becton-Dickinson) and incubated for 24 h at 37°C under aerobic conditions. The anaerobic bacteria were individually grown on HK enriched agar plates (Brain Heart Infusion Agar (BBL, Becton-Dickinson), TSA (BBL, Becton, Dickinson), Yeast extract (BBL, Becton, Dickinson), supplemented with 5 µg/mL of Hemin (Sigma-Aldrich), 0.3 µg/mL of Menadione (Sigma-Aldrich, St. Louis, MO), and 5% defibrinated sheep blood (Microlab, Mexico City)),

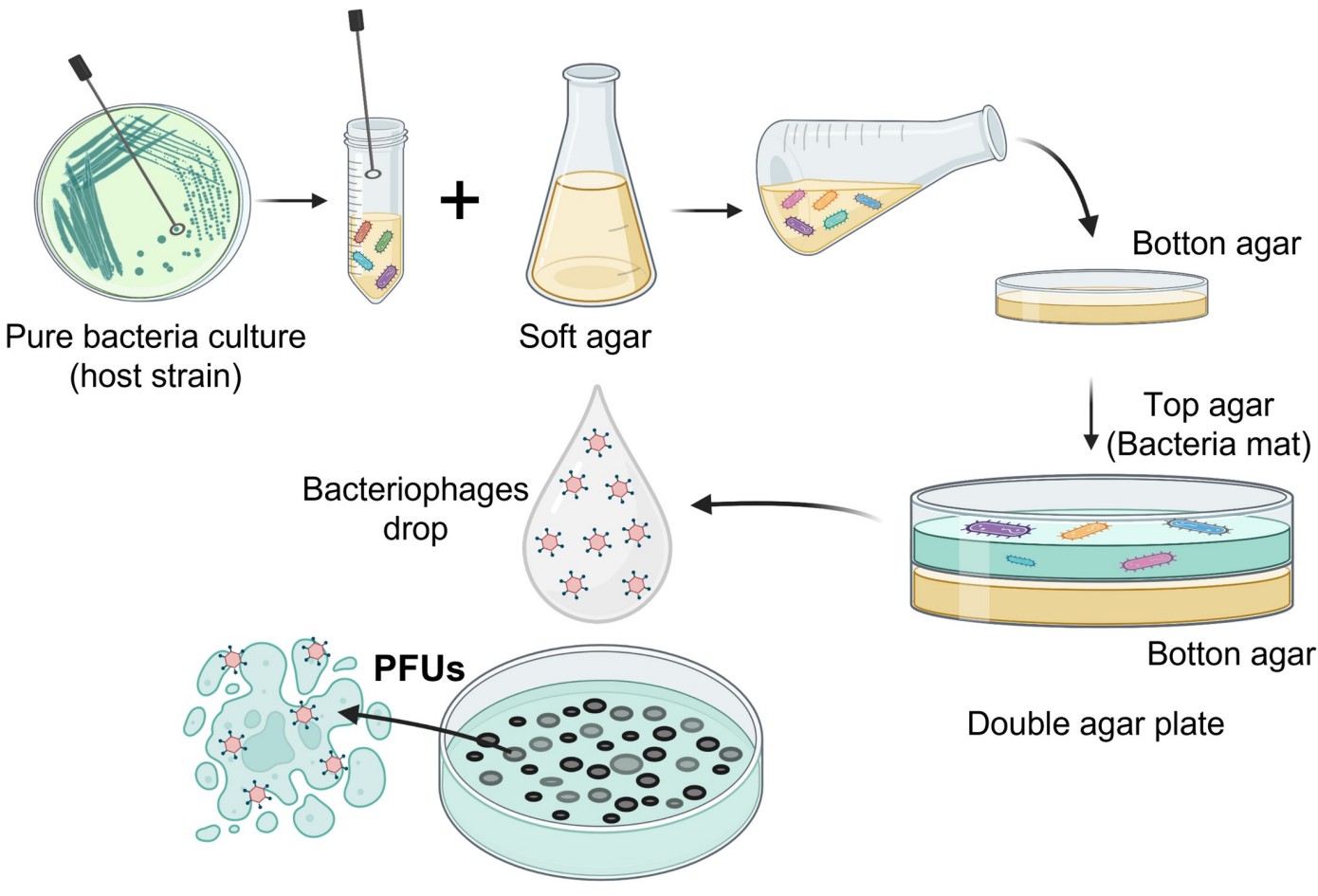

**Fig 1. Schematic figure showing the double-layer plaque assay technique for propagation of bacteriophages.**

**Table 3. ATCC Bacterial species used in the *in vitro* studies.**

| Bacterial strain | ATCC | Gram stain | Atmosphere growth condition |
|---|---|---|---|
| *Actinomyces israelii* | 12102 | + | anaerobic |
| *Aggregatibacter actinomycetemcomitans* serotype b | 43718 | - | anaerobic |
| *Escherichia coli* | 33780 | - | aerobic |
| *Porphyromonas gingivalis* | 33277 | - | anaerobic |
| *Pseudomonas aeruginosa* | 43536 | - | aerobic |
| *Streptococcus mutans* | 25175 | + | anaerobic |
| *Staphylococcus aureus* | 25923 | + | aerobic |
| *Staphylococcus epidermidis* | 14990 | + | aerobic |

and incubated for 5 to 7 days at 35 °C under anaerobic conditions (80% N, 10% $CO_2$ and 10% $N_2$). Pure cultures of each strain were used in the experiments.

After the incubation period, the different bacterial cultures were individually harvested and suspended in tubes containing enriched TSB broth (TSB) or enriched *Mycoplasma* broth (*Mycoplasma* broth (BBL, Becton-Dickinson) added with 5 μg/mL hemin and 0.3 μg/mL menadione), depending on the strains. The optical density (OD) in each tube was adjusted to 1 at λ = 600 nm in a spectrophotometer (BioPhotometer D30, Eppendorf) to obtain a bacterial suspension with 1 x$10^9$ cells/mL.

**ESKAPE bacteria.** The ESKAPE acronym is used to groupsix bacteria species that have been recognized as high virulent nosocomial pathogens, and are associated with resistance to multiple antibiotics [55]. The evaluated ESKAPE bacteria were *Acinetobacter baumannii* (*A. baumannii*), *Klebsiella pneumoniae* (*K. pneumoniae*), *Pseudomonas aeruginosa* (*P. aeruginosa*), *Citrobacter freundii* (*C. freundii*), *and Staphylococcus aureus* (*S. aureus*), obtained from the collection of microbial pathogens of the research unit of the Hospital Juárez de México, HJM [56].

**Bio-aerosols.** Bio-aerosols were simulated preparing a solution with a known concentration of the microorganism and creating an aerosol using a medical grade nebulizer. The set up was based on the filtration system proposed in the standard ASTM: F2101-01 "Standard Test Method for Evaluating the Bacterial Filtration Efficiency (BFE) of Medical Face Mask Materials, Using a Biological Aerosol of *Staphylococcus aureus*" (**Fig 2**).

*Virus aerosol evaluation.* The aerosol solutions were prepared dispersing 600 μL of each surrogate virus in 6 mL of phage buffer (1x$10^8$ PFUs). The solution was placed in the nebulizer beaker. Textile samples of PP with or without the AgCu nanolayer were exposed to 2 min of nebulizer and vacuum pump to attract the phages-contained aerosols, simulating a human breathing system. Agar plates with upper host bacteria lawns were placed in the impactor to recover the phages that could pass through the PP textiles. After the 2 min exposure to the bio-aerosol containing the virus, the agar plates were incubated at 37°C overnight. The number of viable phages that pass through the experimental materials was expressed as the total number of PFUs, which represent the quantity of viruses that are capable of infecting and lysing host cells.

*Bacterial aerosol evaluation.* All bacterial species were taken at 600 μL dispersed in 6 mL of TSB (aerobic strains) or *Mycoplasma* broth (anaerobic strains) culture medium in the nebulizer beaker. The concentration for this test was 1x$10^8$ CFUs/mL. Textile samples of PP (with or without the AgCu nanolayer) were exposed to 2 min of nebulizer and vacuum pump (to attract the bacteria-contained aerosols, simulating a human breathing system). Agar plates were placed in the impactor to recover the bacteria that could pass through the PP (with or without the AgCu nanolayer) fabric. After 2 min exposure to the bio-aerosol containing the bacteria, the agar plates were incubated at 35°C under aerobic or anaerobic conditions. The

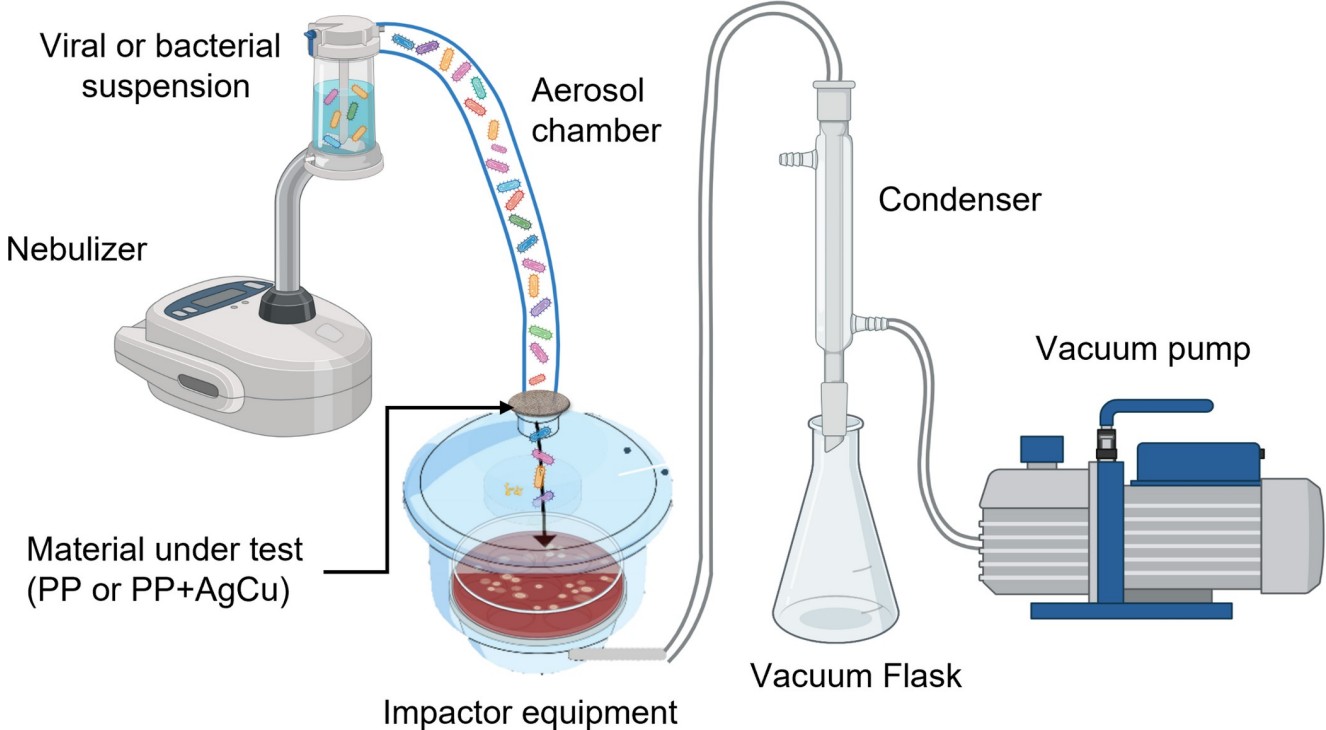

**Fig 2. Schematic figure showing the experimental set-up to simulate bacterial and viral bioaerosols.**

number of viable bacterial cells that could pass through the experimental materials was expressed as the number of CFUs/mL.

*Controls.* To ensure that the viability or infectivity of bacteria and viruses, respectively, was not compromised during propagation in the aerosol medium, a control process was conducted. This involved verifying that the initial concentration presents in the nebulizer beaker closely matched the concentration observed on the agar plates without the presence of any textile (physical barrier). The viral aerosol controls are shown in **S1 Fig**. and the bacterial aerosol controls are shown in **S2 Fig**. In both cases, there is a slight reduction of the microorganism reaching the agar plate but keeping the same logarithmic value.

**On contact tests.** *Surrogate viruses on-contact droplet analysis.* According to the guidelines outlined in ISO 18184, titled "Textiles - Determination of the antiviral activity of textile products", the plaque assay method was employed for the assessment of the on-contact antiviral response. A drop for each phage ($3 \times 10^7$ PFUs in 30 µL) was incubated on 1 cm diameter disc of the AgCu nanolayer and the uncoated PP. The samples were kept at 37˚C during periods of 0.5, 1, 1.5, 2, 4, 6, 12, and 24 h. After the exposure time, the discs were recovered and placed in a sterile tube with 500 µL of phage buffer. Then, phage titer was determined by serial dilutions using the double layer soft agar technique [53, 54, 57]. The results were expressed as the calculation of infectivity titre PFUs/Vial (*Vp*) as indicated in the standard according to the following formula: $V_p = W * C$

where, *Vp* is infectivity titre (PFUs/vial), *W* is infectivity titre per mL (PFUs/mL), *C is* wash-out virus mediation in time 0 (values reported in **S3 Fig**). Then, the viral inactivation

percentage was estimated using the following equation:

$$\text{Viral inactivation (\%)} = [(1 - 10^L)](100\ \%) \tag{2}$$

Where $L$ *is* $\text{Log}_{10}$ (PFUs/mL on PP) - $\text{Log}_{10}$ (PFUs/mL on AgCu).

Given the potential for Ag/Cu ionic leaching from the coated PP during the incubation period, which could impact the bacteria, an additional control experiment was included to ensure that any effects observed on the bacteria host were due to the presence of phages rather than the extract. This control experiment involved applying a drop of culture medium (without phages) onto both the textiles (PP and AgCu nanolayer) and allowing it to incubate for 24 hours. Following this incubation period, the samples were titrated and subsequently seeded on agar plates containing the respective host bacteria (*P. aeruginosa S33* and *E. coli rfaB*). No bacteria lysis was observed due to the extracts obtained from uncoated or coated PP. The figures illustrating the results of this control experiment can be found in the supporting information file (**S4 Fig**).

*Bacterial on-contact droplet analysis*. Droplets of 40 μL ($4\times10^7$ CFUs/mL) of each bacterial strain were individually placed on PP discs (1 cm diameter) with or without the AgCu nanolayer and incubated at 35°C for 24 h under aerobic or anaerobic conditions. After incubation, the discs were recovered and placed in a sterile tube with 500 μL of culture medium. Four-serial dilutions (1:100) were made and cultured by pipetting 5 μL of each of the bacterial dilutions into agar Petri dishes and incubated at 35°C under aerobic (24 h) or anaerobic (5 days) conditions depending on the bacteria strain. The percentage of bacterial inhibition was expressed as:

$$\text{Bacterial inhibition (\%)} = [(1 - 10^L)](100\ \%) \tag{3}$$

Where $L$ *is* $\text{Log}_{10}$ (CFUs/mL on PP) - $\text{Log}_{10}$ (CFUs/mL on AgCu)

*ESKAPE bacteria*. To investigate the antibacterial activity of the single-sided AgCu coated PP against ESKAPE bacteria, the same methodology as described in reference [38] was employed. However, in this case, the samples were tested only for the highest concentration of $10^5$ CFU/50 μL. After the incubation time, the ESKAPE bacteria were eluted from the discs using saline solution. Aliquots obtained from the contact assays with ESKAPE bacteria were appropriately diluted, and viable counts were conducted by plating on Luria Bertani agar. Following incubation, the colonies were counted and reported as colony-forming units (CFU), and the average CFU was compared before and after the contact time. The bacterial inhibition percentage was estimated based on the CFUs obtained for the AgCu nanolayer and the PP, utilizing Eq (3).

**Statistical analysis.** The biological experiments were conducted in triplicate using three independent samples of each group of samples. The results are expressed as the mean values ± SEM (standard error of the mean). The statistical significance was determined by paired T-student test.

## Results

### AgCu nanolayer

The AgCu nanolayer's composition and uniformity were assessed using SEM-EDS (Jeol 7600). For this, pieces of 1 cm diameter were cut randomly from the 18 cm wide PP roll. **Fig 3** demonstrates the uniform deposition of the film on the PP fibers. The EDS analysis revealed a composition of 39 ± 7 at.% Ag and 61 ± 7 at.% Cu. The slightly higher Cu content compared to our previous report can be attributed to the different target configuration used. The reported

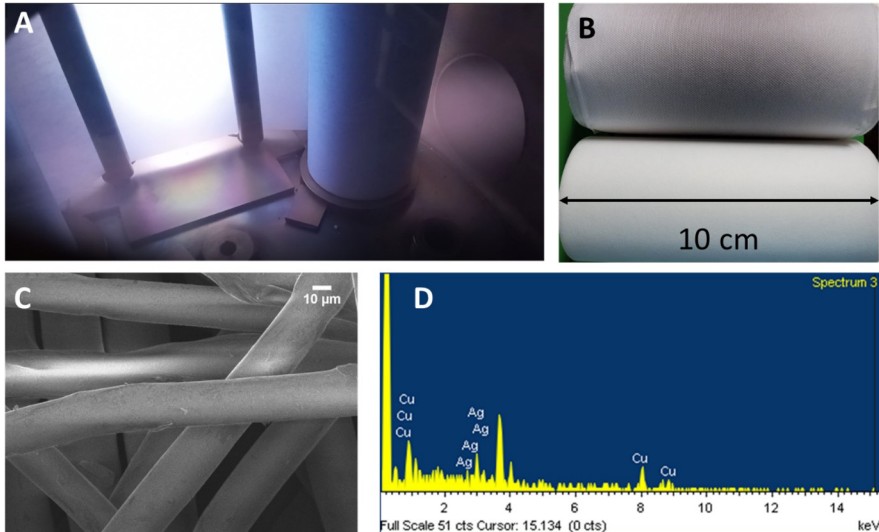

**Fig 3. AgCu nanolayer. (A)** Interior of the vacuum chamber during the roll-to-roll deposition, **(B)** coated and uncoated polypropylene rolls, **(C)** SEM image of the PP fibers coated with the AgCu nanolayer **(D)** representative EDS spectra.

values represent the average measurements obtained from different deposition runs (3) in randomly chosen pieces (3–4) of the coated PP. The low standard deviation indicates uniform coating and successful repeatability.

## Filtration efficiency (NaCl)

Results of the filtration efficiency of sodium chloride particles for three different particle sizes (0.3, 0.5, and 1 μm) are shown in **Table 4**. The results indicate no statistically significant differences between the filtration efficiency estimated for polypropylene [45] fabric and the AgCu-coated PP (SakCu®). This test indicates that the application of the AgCu nanolayer does not cause significant changes in the size of the pores and fiber diameter of the PP fabric, resulting in similar filtration efficiencies. These findings suggest that any variation in viral infectivity or bacterial growth observed in the bio-aerosol experiments can be attributed to the antimicrobial properties of the AgCu nanolayer rather than alterations in filtration efficiency.

## Bioaerosols

**Surrogate viruses.** **Fig 4** shows the evaluation of the antiviral properties of the SakCu® nanolayer by the simulated filtration system using aerosols containing surrogate's virus that encountered the polypropylene fabric, with and without the AgCu nanolayer. A significantly reduced number of PFUs were detected when the viral aerosol passes through the textile with the AgCu nanolayer, except for the PhiX174 phage. The percentage of viral aerosol inactivation

**Table 4. NaCl filtration efficiency (%) in the uncoated-PP [45] and AgCu-coated PP (AgCu).**

| | | | |
|---|---|---|---|
| PP | 30.5±8.3 | 61.1 ±6.6 | 72.1 ± 0.6 |
| AgCu | 28.1±2.6 | 56.9 ±3.7 | 66.4 ±2.8 |

SEM = Standard error of the mean

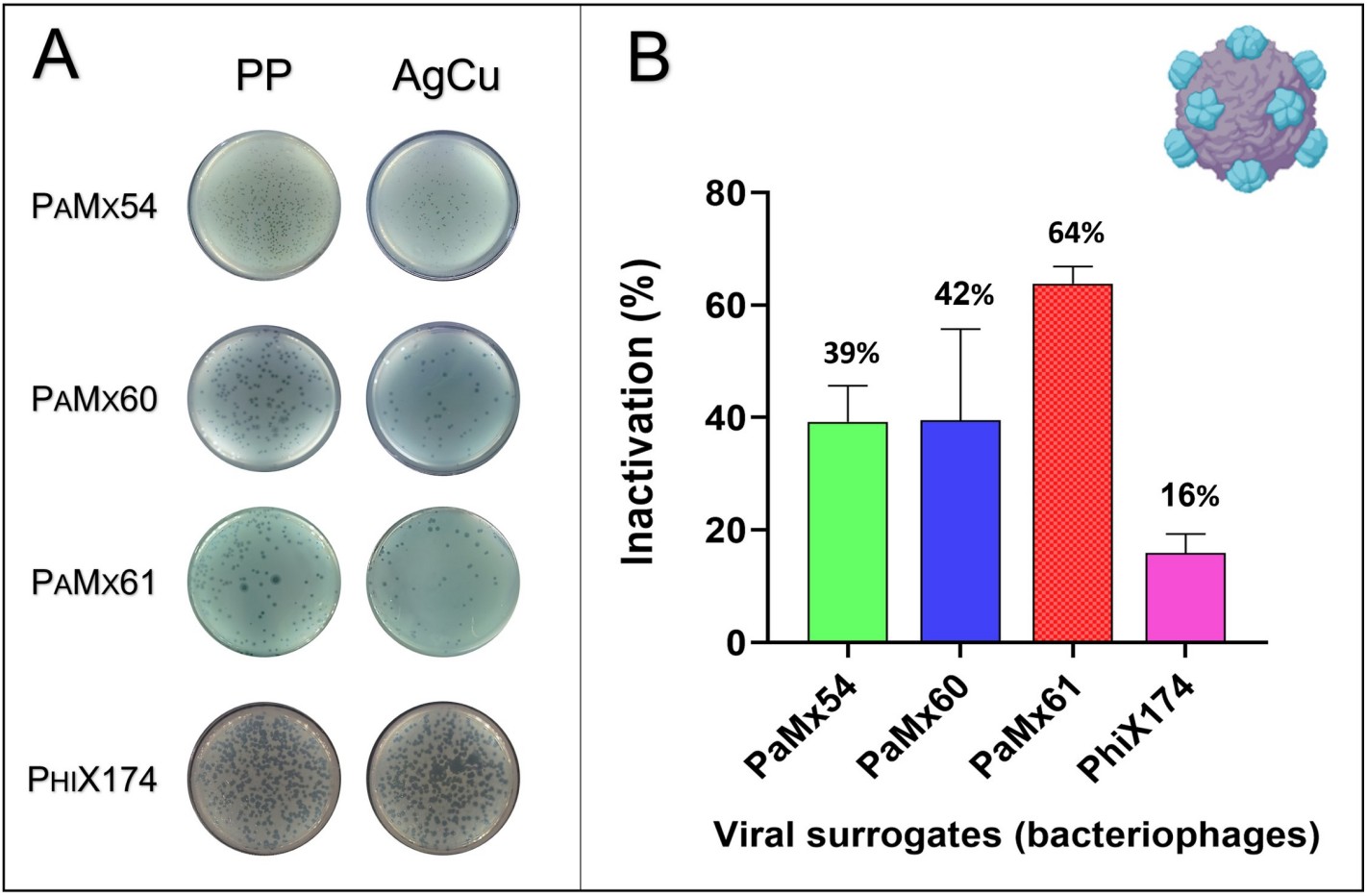

**Fig 4. Viral aerosol evaluation. (A)** Representative agar plates in which the PFUs of the different bacteriophages after the aerosol assay were counted. **(B)** Percentage of viral aerosol inactivation of the AgCu nanolayer vs. PP using Eq 2.

was between 16 to 64%. The PaMx54, PaMx60, and PaMx61 bacteriophages were the most sensitive to contact with the AgCu nanolayer.

**ATCC bacteria.**    The antibacterial evaluation in the simulated filtration system is presented in **Fig 5**. The results show that when the aerosol passed through the textile with the AgCu nanolayer, the number of viable bacteria was significantly reduced in all bacterial strains. The bacterial inhibition percentage achieved when in contact with the AgCu nanolayer exceeded 70% for both aerobic and anaerobic bacteria groups. Specifically, the inhibition ranged from 70% to 92% for aerobic bacteria and 72% to 95% for anaerobic bacteria.

### On-contact

**Surrogate viruses.**    The evaluation of viral surrogates in contact with the AgCu nanolayer demonstrated complete inactivation (100%) of RNA phages and nearly 100% inactivation of DNA phages. **Fig 6A** illustrates that the phages exposed to uncoated PP retained their infective capacity after 24 hours. However, the phages exposed to the AgCu nanolayer exhibited a significant reduction in infectivity after 30 minutes of exposure, and by 2 and 4 hours, they completely lost their infective capacity. **Fig 6B** shows the viral inactivation percentages of the bacteriophages at different incubation times vs. PP. The RNA phages (PaMx54, 60, and 61)

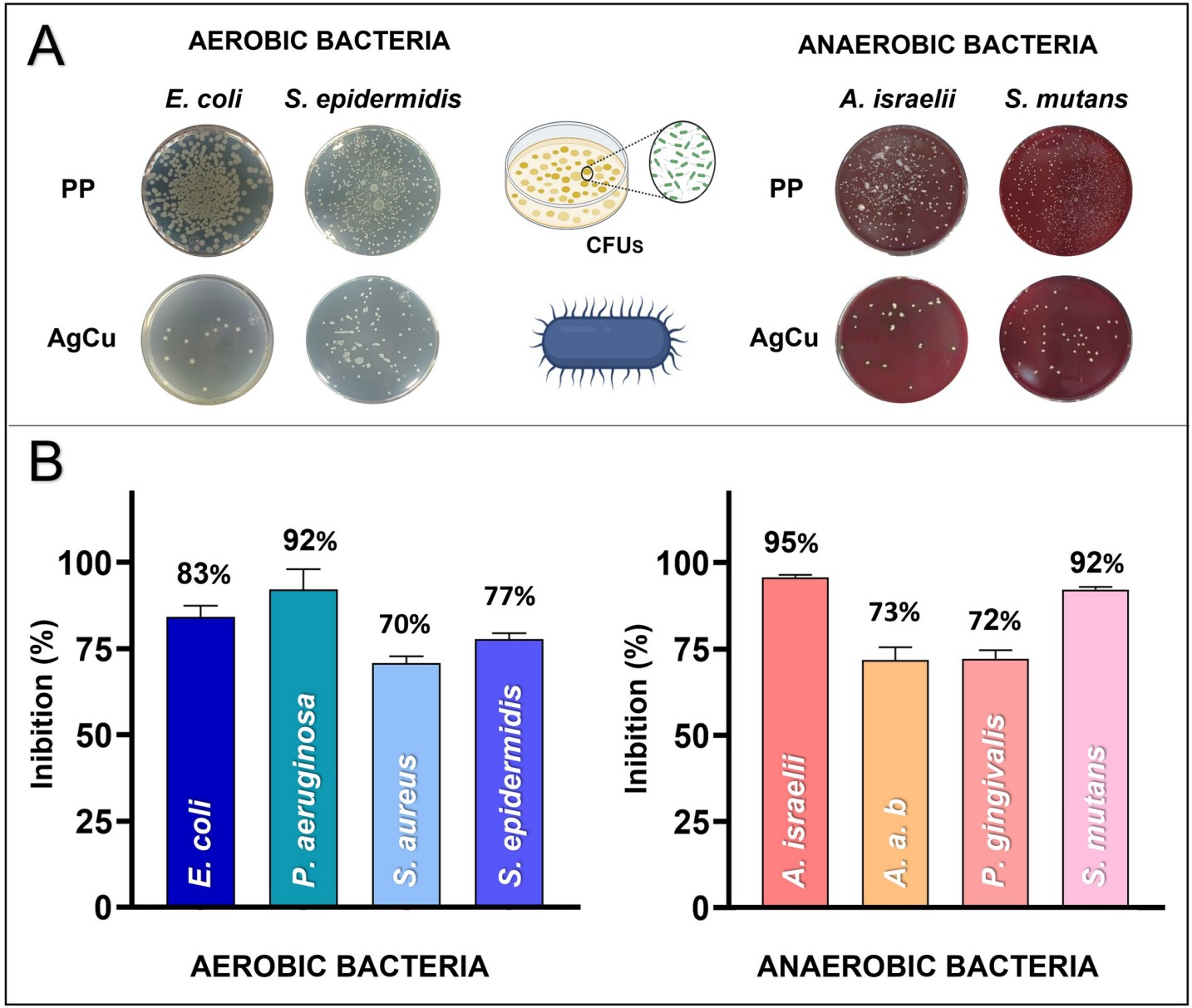

**Fig 5. Bacterial aerosol evaluation. (A)** Representative agar plates in which the CFUs of the different aerobic and anaerobic bacteria after the bio-aerosol assay were counted. **(B)** Percentage of bacterial aerosol inhibition of the AgCu nanolayer vs. PP using Eq 3.

were 100% inactivated between 2 and 4 h. And the DNA phage (PhiX174) was 92% inactivated after 12 h of contact with the Ag nanolayer.

*SARS-CoV-2.* **S5 Fig** confirms the inactivation of the SARS-CoV-2 as a function of the contact time with the AgCu nanolayer deposited on one side of the PP in a similar trend as observed in the previous work [46]. The genetic material is no longer detected by the PCR technique after 8h in contact with SakCu®, meanwhile, on the PP, the RNA is still detectable.

**Antibacterial evaluation.** The bacterial on-contact droplet analysis revealed that the AgCu nanolayer exhibited significant antibacterial efficacy against all tested strains, with particularly notable effectiveness against anaerobic bacteria (**Fig 7**). For the aerobic bacteria, the inhibition percentage ranged from 80% to 99%. Similarly, all anaerobic species demonstrated

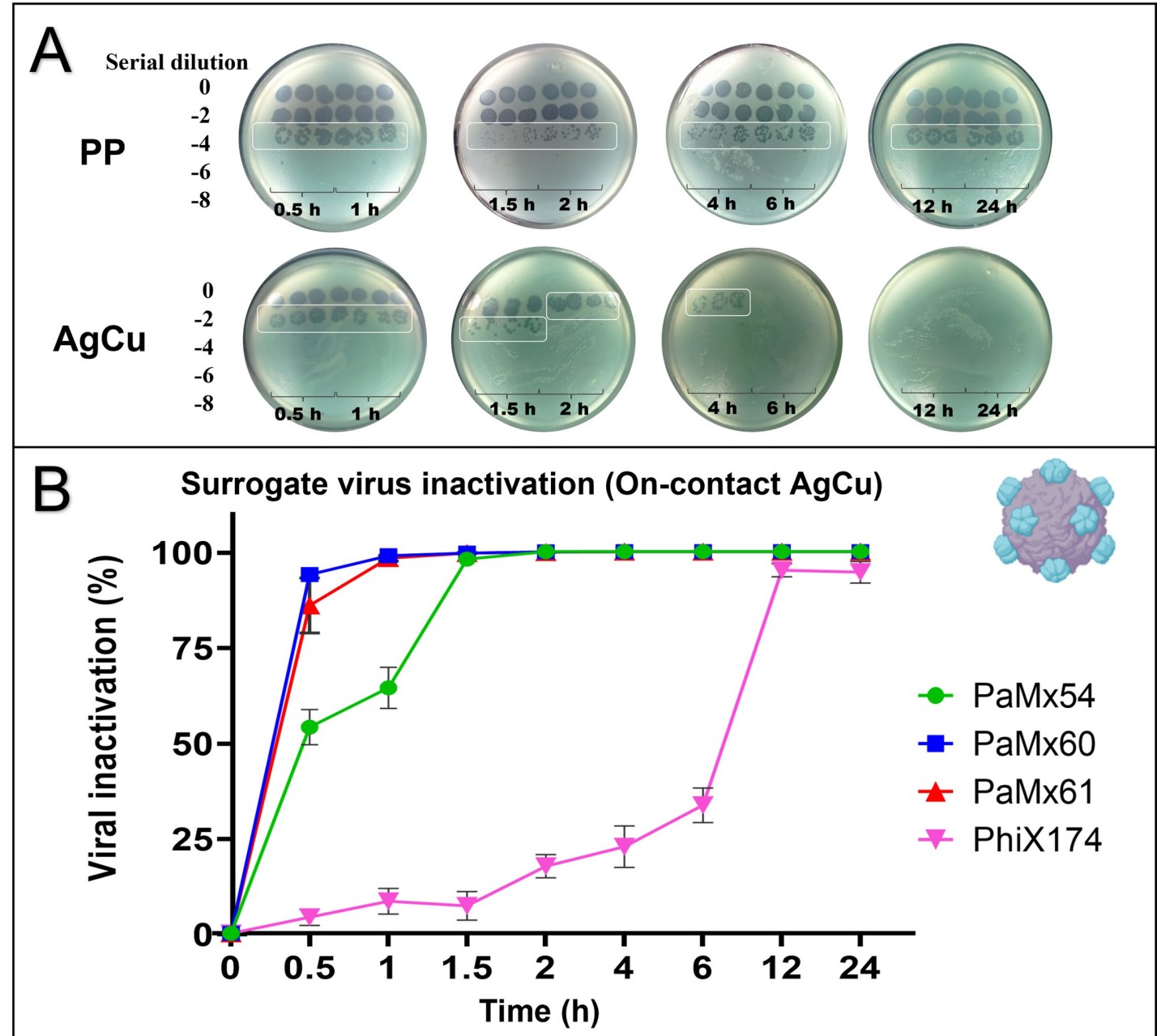

**Fig 6. Surrogate virus inactivation. (A)** Representative figures of PFU/mL determination in on-contact droplet assays after different incubation times (0.5–24 h) with or without the AgCu nanolayer. It is shown results spotting serial dilution of PaMx60 on its bacteria host strain. Each experiment was performed in triplicate and spotting three times each serial dilution. **(B)** Time evolution of the viral inactivation percentage due to contact with the AgCu nanolayer.

inhibition rates exceeding 92% (Fig 7). Additional information, including the figures displaying the CFUs dilutions of aerobic and anaerobic bacteria in the droplet assay, can be found in the supplementary material (S6 Fig).

**ESKAPE bacteria.** Fig 8 demonstrates that the AgCu nanolayer effectively suppressed the growth of *C. freundii*, *K. pneumoniae*, and *S. aureus* strains after two hours of contact. However, the inhibition of *A. baumannii* and *P. aeruginosa* strains occurred gradually with increasing contact time. The inhibition percentages were calculated using Eq (3). For *P. aeruginosa*,

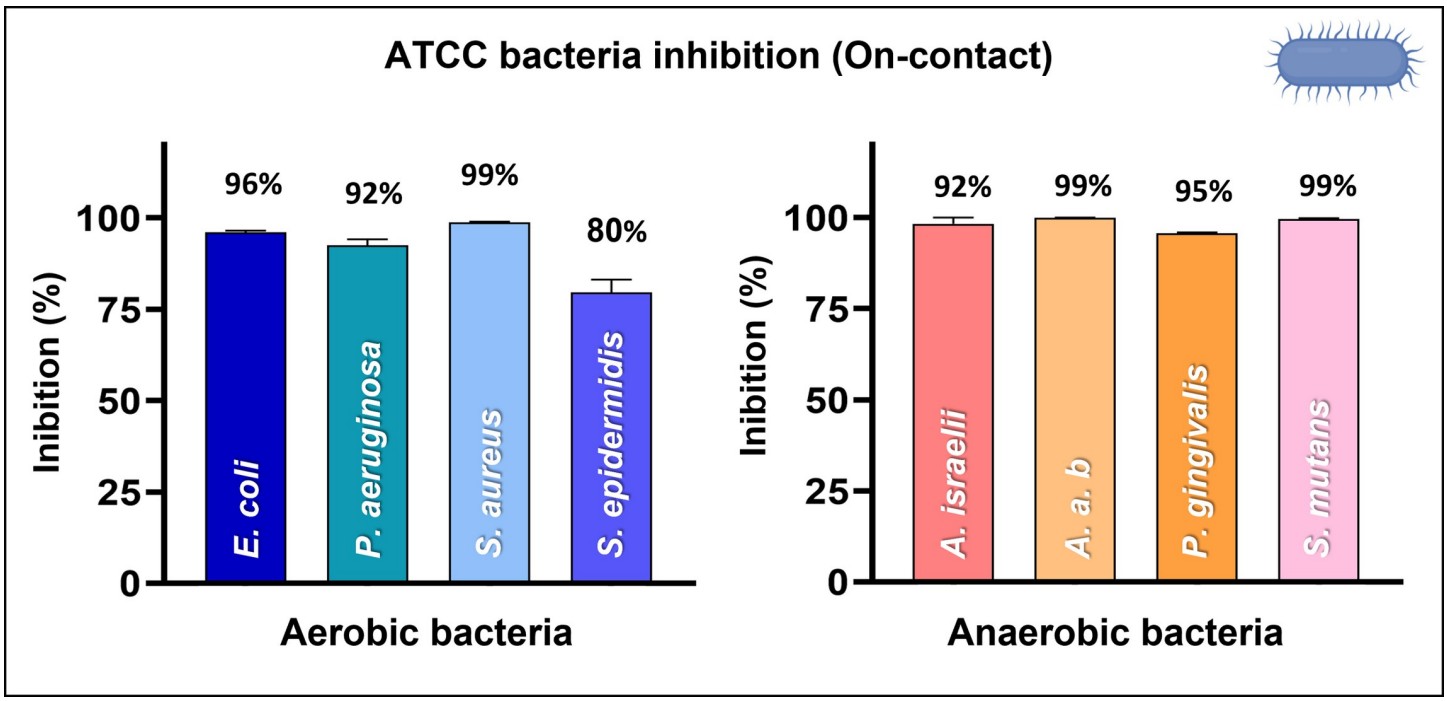

**Fig 7. Bacteria on-contact droplet analysis.** The inhibition percentage due to contact with the AgCu nanolayer vs. PP using Eq 3. Each experiment was performed in triplicate and spotting three times each serial dilution.

inhibition percentages of 15%, 39.4%, 79.1%, and 100% were observed, corresponding to 2, 4, 8, and 24 hours of contact, respectively. In the case of *A. baumannii*, inhibition percentages of 9%, 24%, 86.5%, and 100% were detected for 2, 4, 8, and 24 hours of contact, respectively. These results are relevant because ESKAPE pathogens are known as drug-resistant bacteria and are currently the most important cause of hospital-acquired infections. Thus, the AgCu nanolayer could be an effective method to inhibit biofilm formation in personal protective equipment and other medical devices.

## Discussion

In this study, we evaluate the capacity of the AgCu nanolayer (SakCu®) deposited by the magnetron sputtering technique on one side of polypropylene fabric to inactivate respiratory surrogate viruses, nosocomial and oral bacteria in aerosol-borne and on-contact.

The AgCu nanolayer was uniformly deposited on the PP fibers, one of the main advantages of using the magnetron sputtering technique is the possibility of depositing a homogeneous nanometer-sized coating on different surfaces [58]. SEM showed the uniformity of the nano-coating on the PP fibers, and EDS analysis confirmed the presence and percentage of Ag and Cu elements in the composition.

The antimicrobial evaluation of the AgCu nanolayer is a significant contribution to this study. It involved the assessment of its effectiveness against various microorganisms, including SARS-CoV-2, surrogate viruses of RNA (PaMx54, PaMx60, PaMx61), and DNA (PhiX174), which represent different surrogate models for respiratory viruses like norovirus (NoV) [45], influenza virus (H1N1) [59], and others [60]. Additionally, twelve nosocomial bacteria, comprising aerobic and anaerobic respiratory and oral pathogens, were included in the evaluation.

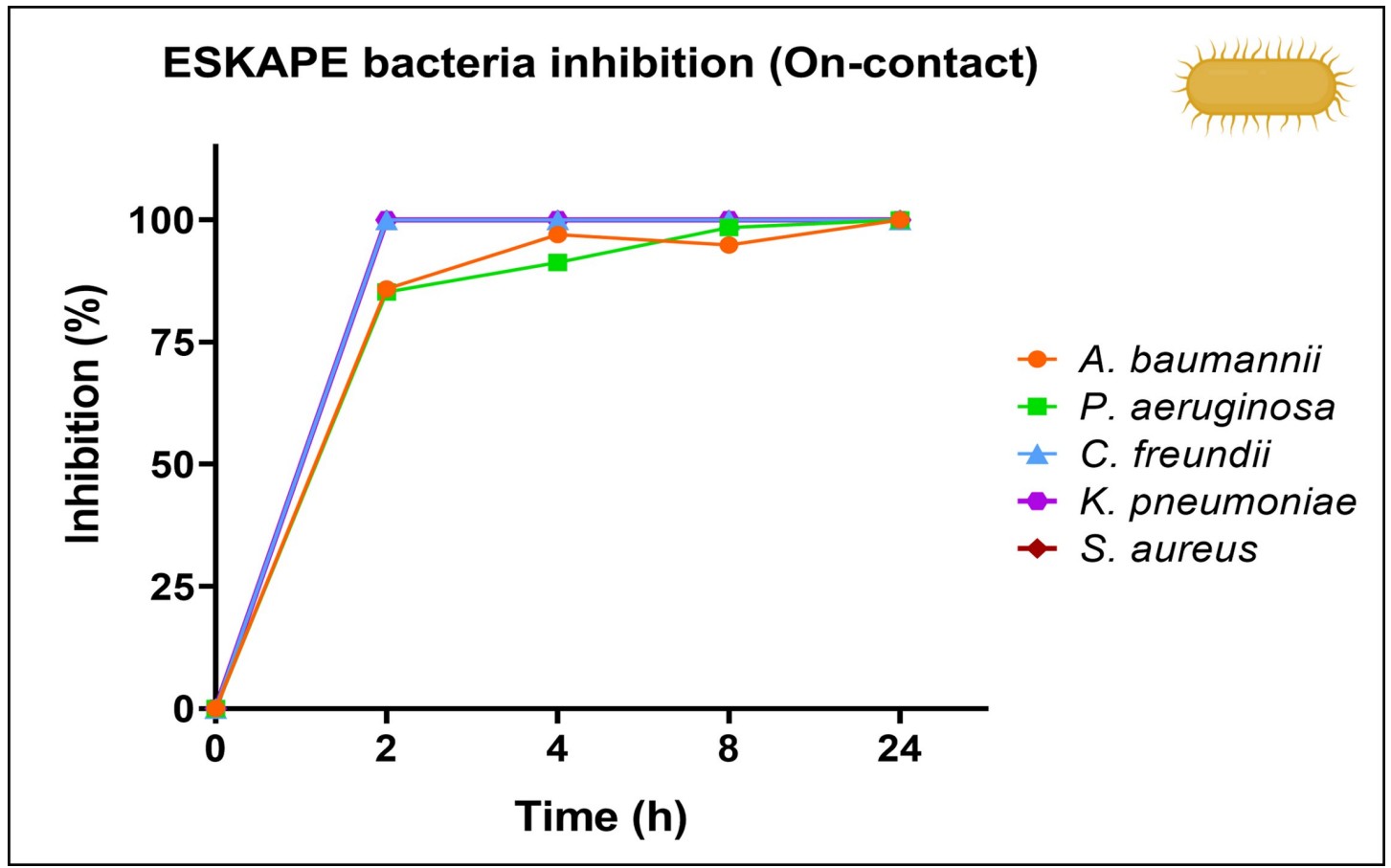

**Fig 8. ESKAPE bacteria on-contact droplet analysis.** Time evolution of the bacterial inhibition percentage due to contact with the AgCu nanolayer vs. PP using Eq 3.

This comprehensive assessment covered two different methodologies: aerosol exposure for 2 minutes and direct on-contact droplet tests at various time intervals ranging from 0 to 24 hours.

Our results demonstrate that AgCu nanolayer was more effective against the ssRNA bacteriophages (PaMx54, PaMx60, and PaMx61) than to the DNA phage (PhiX174) since complete inactivation was achieved in 2–4 hours while 12 hours were required to inactivate the DNA virus. Similarly, in our previous study, the DNA virus Human Papillomavirus (HPV) subtypes types 16 and18 were not sensitive to the AgCu nanofilm [46]. Studies have reported variations in the susceptibility of viruses to disinfectants or virucides based on their characteristics, such as DNA vs. RNA or enveloped vs. non-enveloped viruses. However, the underlying reasons for these differences remain largely unexplored. For instance, Sagripanti et al. [61] investigated the virucidal activity of $Cu^{2+}$ ions against a range of enveloped and non-enveloped DNA and RNA viruses. Their findings revealed that $Cu^{2+}$ ions were highly effective against RNA viruses but had limited efficacy against DNA viruses. Nevertheless, a mixture of copper and peroxide could eliminate all five virus types studied. The authors proposed that this could be attributed to the larger toxicity of $Cu^+$ ions resulting from the interaction with the peroxide or due to the generation of reactive oxygen species [61].

More recently, Soliman et al. [62]. reported that Cu ions failed to inactivate the DNA virus Phi X174 within a pH range of 5–8, whereas the RNA virus, MS2, showed significant reduction in infectivity. In contrast, silver (Ag) ions demonstrated effectiveness against both types of

viruses, with the degree of efficacy dependent on pH levels. A possible explanation for these results lies in the electrostatic interactions between the metal ions and the amino acids comprising the viral capsid, a phenomenon closely tied to the unique characteristics of each virus [62]. Nevertheless, Cheng et al. compared the inactivation of MS2 and PhiX174 by nanoscale zero-valent iron concluding that both viruses suffer capsid damage but the nucleic acid of MS2 (RNA) was completely degraded in 240 min, while the DNA in PhiX174 was simply more resistant [63].

Furthermore, the AgCu nanolayer achieved and maintained 100% inhibition of SARS-CoV-2 within a time frame of less than 24 hours upon contact with the nanofilm deposited on one side of the PP fabric (**S5 Fig**). This highlights the potent antiviral activity of the AgCu nanolayer. An additional advantage of using this AgCu metallic coating is that it does not require photocatalytic activation for the inhibition of viruses, unlike $TiO_2$, which necessitates photocatalytic activation for its antiviral effects [38].

In addition to the demonstrated viricidal effect, the AgCu nanolayer also exhibits strong antimicrobial properties against bacteria. These properties were thoroughly evaluated in our study, and the results clearly demonstrated significant inhibition of both aerobic and anaerobic bacteria. The effectiveness of the AgCu nanolayer was observed across different methodologies and interaction conditions, including aerosol exposure and on-contact (drops). This comprehensive evaluation sets our study apart from others that often concentrate on testing the bactericidal capacity of Ag, Cu, or other materials against only one or two specific bacteria.

For instance, some studies have evaluated Cu-Ag coatings solely against *S. aureus* [27, 31], while in another work, a membrane loaded with AgCu-NPs on PP was tested against two bacteria, namely *E. coli* and *S. aureus* [64]. In contrast, our study aimed to assess the antibacterial activity of the AgCu nanolayer against a total of twelve Gram-positive and Gram-negative respiratory and oral nosocomial bacteria. This broad evaluation provides a comprehensive understanding of the nanolayer's efficacy against different bacterial strains.

The antibacterial properties of silver (Ag) and copper (Cu) have long been recognized, and their effectiveness as antimicrobial agents continues to be studied. These elements, whether used individually or in combination as nanoparticles or coatings, have demonstrated bactericidal effects and exhibit antimicrobial synergy [65–71]. For instance, previous studies have reported significant inhibition of various bacterial species, including *E. coli*, *S. aureus*, *A. baumannii*, *and Bacillus subtilis*, when exposed to nanomaterials composed of Ag and Cu [67, 70]. Moreover, several studies have shown that Ag and Cu metals have a more effective bactericidal effect than other metals; such as Al, Zr, and Ti against *E. coli* [72] and *P. aeruginosa* [73].

Besides, studies have shown that Ag nanoparticles (Ag-NPs) and Cu nanoparticles (Cu-NPs) have demonstrated higher inhibition percentages against oral bacteria compared to nanoparticles derived from bismuth or zirconium [74–77]. This highlights the superior antimicrobial properties of Ag and Cu in the context of oral bacteria. In this study, we observed that oral anaerobic species displayed a high sensitivity to the AgCu nanolayer. This finding is intriguing because previous research has often reported diminished antibacterial activity under anaerobic conditions [78, 79]. The rationale behind this phenomenon lies in the absence of oxygen during anaerobic conditions, which limits the availability of metallic ions ($Cu^{2+}$ and $Ag^+$), consequently reducing their toxicity towards microorganisms [74, 80]. However, it's worth noting that under anaerobic conditions, metals can still undergo corrosion in aqueous solutions through an alternative mechanism involving the oxidation of the metal into a metal hydroxide.

Although there are still questions about the mechanism of action of metallic nanomaterials on bacteria and viruses, it is known that in humid conditions, they can release metal ions, causing oxidative stress, cell membrane and genetic material damage of viruses and bacteria

[40]. Previously, the possible mechanism of action of the AgCu nanolayer was studied by quantum chemistry calculations showing that the addition of Ag and Cu makes the polymeric fiber a better electron acceptor, this can promote the oxidation of the phospholipids present in both, virus and bacterial membranes, and the rupture at the membrane exposes and damages virus´s genetic material [46]. Indeed, one of the key antimicrobial mechanisms exhibited by Cu-containing surfaces is known as "contact killing." This mechanism involves the rapid killing of bacteria, yeasts, and viruses upon contact with copper metal surfaces. The effectiveness of this process is primarily attributed to the actions of copper ions. Cu ions induce several killing processes that result in the demise of microorganisms. One of these processes involves cell damage, where the presence of copper ions leads to structural and functional impairment of microbial cells. Additionally, the generation of reactive oxygen species (ROS) is triggered by Cu ions, further contributing to the destruction of microorganisms. The accumulation of copper and other stress factors can cause rupture of the cell membrane, resulting in the loss of membrane potential and leakage of cytoplasmic content. Furthermore, Cu ions can degrade both genomic and plasmid DNA, interfering with essential cellular functions. These mechanisms, along with others, collectively contribute to the potent antimicrobial properties of Cu-containing surfaces, making them highly effective in killing a wide range of microorganisms upon contact [42, 44, 81, 82].

Even if we have not systematically compared the antimicrobial effect of Ag and Cu versus the AgCu nanolayer, the great results obtained are indicative of a synergy between both metals. One advantage of the proposed method is the production of a continuous nanolayer, there are not individual nanoparticles that could be free and transport into the living organisms, where cytotoxic or inflammatory response could be triggered. The adhesion between the AgCu nanolayer and the PP was previously evaluated using high air fluxes or a tearing test [46].

## Conclusions

The AgCu nanolayer, specifically SakCu®, which is deposited on polypropylene fabric using magnetron sputtering, has demonstrated remarkable antiviral potential against respiratory viruses, including SARS-CoV-2 and surrogate viruses. The nanolayer exhibits an exponential reduction in viral load within a short timeframe of 95.6% inhibition in 8 hours for SARS-CoV-2, and nearly 100% inactivation of the RNA surrogate virus PMAx60, highlighting its efficacy in combating airborne respiratory viruses. Additionally, the AgCu nanolayer displays excellent antimicrobial properties against a wide range of pathogenic aerobic and anaerobic bacteria.

Given its potent antiviral and antimicrobial capabilities, the AgCu nanolayer holds great promise for applications in personal protective equipment (PPE) such as face masks. By incorporating the AgCu nanolayer into PPE, the transmission of aerosol-borne pathogenic bacteria and viruses, including SARS-CoV-2, can be significantly reduced, and prevented. This suggests that the AgCu nanolayer has the potential to enhance the protective capabilities of PPE and contribute to the overall reduction of infectious diseases.

Furthermore, the observed antibacterial properties of the AgCu nanolayer against both aerobic and anaerobic pathogens, including antibiotic-resistant strains, make it a valuable tool in reducing healthcare-associated infections. Implementing the AgCu nanolayer in healthcare settings could serve as an effective measure to mitigate the spread of bacteria and contribute to infection control [83].

## Supporting information

**S1 Table. Viral and bacterial aerosol supporting information.**
(DOCX)

**S1 Fig. Viral aerosol control.** Viral concentration (PFUs) quantified from the agar plates is almost the same as the initial viral suspension.
(TIF)

**S2 Fig. Bacteria aerosol control.** Bacteria concentration (CFUs) quantified from the agar plates is almost the same as the initial bacterial suspension.
(TIF)

**S3 Fig. Infectivity titre supporting information.**
(TIF)

**S4 Fig. Control experiment.** Control experiment to illustrate that the extracts obtained from the uncoated and coated PP do not affect the bacteria host from the surrogate viruses. A. Bacterial mat using only culture media. B. Bacterial mat adding the extracts. C Representative image of the bacteria lysis using the surrogate viruses.
(TIF)

**S5 Fig. Viral inactivation exponential evolution of SARS-CoV-2 against AgCu nanolayer.**
(TIF)

**S6 Fig. Representative figures of bacterial growth on agar plates after 24 h on-contact with or without the AgCu nanolayer.**
(TIF)

## Author Contributions

**Conceptualization:** Argelia Almaguer-Flores, Sandra E. Rodil.

**Formal analysis:** Lorena Reyes-Carmona, Omar A. Sepúlveda-Robles, Argelia Almaguer-Flores, Juan Manuel Bello-Lopez, Sandra E. Rodil.

**Funding acquisition:** Argelia Almaguer-Flores, Sandra E. Rodil.

**Investigation:** Lorena Reyes-Carmona, Sandra E. Rodil.

**Methodology:** Lorena Reyes-Carmona, Omar A. Sepúlveda-Robles, Argelia Almaguer-Flores, Juan Manuel Bello-Lopez, Carlos Ramos-Vilchis, Sandra E. Rodil.

**Project administration:** Sandra E. Rodil.

**Supervision:** Omar A. Sepúlveda-Robles, Argelia Almaguer-Flores, Sandra E. Rodil.

**Visualization:** Argelia Almaguer-Flores, Sandra E. Rodil.

**Writing – original draft:** Lorena Reyes-Carmona, Argelia Almaguer-Flores, Sandra E. Rodil.

**Writing – review & editing:** Lorena Reyes-Carmona, Argelia Almaguer-Flores, Sandra E. Rodil.

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
