## [Decision Letter · Decision Letter 0]

2 Oct 2023

PONE-D-23-20447Antimicrobial activity of Silver-Copper coating against aerosols containing surrogate respiratory viruses and bacteriaPLOS ONE

Dear Dr. RODIL,

Thank you for submitting your manuscript to PLOS ONE. After careful consideration, we feel that it has merit but does not fully meet PLOS ONE’s publication criteria as it currently stands. Therefore, we invite you to submit a revised version of the manuscript that addresses the points raised during the review process.

We look forward to receiving your revised manuscript.

Kind regards,

Amitava Mukherjee, ME, Ph.D.

Academic Editor

PLOS ONE

Journal Requirements:

3. Please expand the acronyms DGAPA-PAPII and  CONAHCYT (as indicated in your financial disclosure) so that it states the name of your funders in full.

"SE Rodil acknowledges support from SECTEI 096/2020 (https://www.sectei.cdmx.gob.mx/). A. Almaguer-Flores acknowledges support from DGAPA-PAPIIT IT201121 (https://dgapa.unam.mx/). L. Reyes-Carmona thanks CONAHCYT (https://conahcyt.mx/) for the PhD scholarship (CVU 917708).  All authors are grateful for the technical support from  Dr. Gina-Prado and L. Cruz-Fonseca. O. Sepulveda thanks Dr. Dimitris Geogellis for providing the E. coli rfaB bacterial strain."

6. We note that Figures 1, 2, 4, 5, 6, S4 and S5 in your submission contain copyrighted images. All PLOS content is published under the Creative Commons Attribution License (CC BY 4.0), which means that the manuscript, images, and Supporting Information files will be freely available online, and any third party is permitted to access, download, copy, distribute, and use these materials in any way, even commercially, with proper attribution. For more information, see our copyright guidelines: http://journals.plos.org/plosone/s/licenses-and-copyright.

a. You may seek permission from the original copyright holder of Figures 1, 2, 4, 5, 6, S4 and S5 to publish the content specifically under the CC BY 4.0 license. 

Reviewers' comments:

Reviewer's Responses to Questions

**Comments to the Author**

1. Is the manuscript technically sound, and do the data support the conclusions?

Reviewer #1: Yes

2. Has the statistical analysis been performed appropriately and rigorously? 

Reviewer #1: Yes

3. Have the authors made all data underlying the findings in their manuscript fully available?

Reviewer #1: Yes

4. Is the manuscript presented in an intelligible fashion and written in standard English?

Reviewer #1: Yes

5. Review Comments to the Author

Reviewer #1: Introduction: The introduction is well structured.

Add appropriate citation in line 72 where different viruses are claimed to be transmitted mainly through aerosols.

Materials and Methods:

1. The methodology to create the nanocoating is well described. It would additionally be good to add the thickness of the PP used.

2. It would be good to understand how deep within the PP does the coating deposit. Hence SEM and EDS at multiple thicknesses till no AgCu is detected within the fiber would be useful but is optional.

Results:

1. In line 315 describe in which locations and in how many locations was the EDS collected.

2. The x-axis for figures 5 and 7 incorrectly spell inhibition as ”inibition”.

Discussion:

The discussion explains the key results and elaborates well, on the different mechanisms of antibacterial materials.

1. A potential mechanism for less activity of PhiX174 virus should be elaborated based on understanding the cited work (citation 54 and 56).

2. Line 451-4522 mentions ”The oral anaerobic species exhibited a higher sensitivity to the AgCu nanolayer, which is an interesting finding”. It should be clarified what makes this observation interesting. Example: Is it the first time this has been demonstrated?

Conclusion: Conclusion should be more quantitative to describe key findings. For example the statement ”It exhibits an exponential reduction in viral load within a short timeframe, highlighting its efficacy in combating airborne respiratory viruses” should mention the timeframe quantitatively (can use any specific virus studied or mention the range).

6. PLOS authors have the option to publish the peer review history of their article (what does this mean?). If published, this will include your full peer review and any attached files.

Reviewer #1: No

---

## [Author Response · Author response to Decision Letter 0]

9 Oct 2023

I would like to thank you for the careful revision of our manuscript entitled “Antimicrobial activity of Silver-Copper coating against aerosols containing surrogate respiratory viruses and bacteria” 

Moreover, we appreciated the overview given since you have identified the critical points of the work.

INTRODUCTION

1. Minor Revision: Add appropriate citation in line 72 where different viruses are claimed to be transmitted mainly through aerosols.

R: Thanks for this vital suggestion. The modified text with references is like this:

The bio-aerosols are the main source of transmission of respiratory microorganisms, including the severe acute respiratory syndrome coronavirus-2 (SARS-CoV-2) [2-5], coronaviruses (SARS and MERS), rubeola virus (measles) [6], influenza virus [7, 8] or varicella-zoster virus (chickenpox) [9] which can be transmitted in hospitals [10, 11] or even in the environment [12].

2. Tang JW, Wilson P, Shetty N, Noakes CJ. Aerosol-Transmitted Infections-a New Consideration for Public Health and Infection Control Teams. Curr Treat Options Infect Dis. 2015;7(3):176-201. Epub 2015/01/01. doi: 10.1007/s40506-015-0057-1. PubMed PMID: 32226323; PubMed Central PMCID: PMCPMC7100085.

3. World Health O. Transmission of SARS-CoV-2: implications for infection prevention precautions: scientific brief. Geneva: 2020 Contract No.: 2020.3.

4. Wang CC, Prather KA, Sznitman J, Jimenez JL, Lakdawala SS, Tufekci Z, et al. Airborne transmission of respiratory viruses. Science. 2021;373(6558):eabd9149. doi: doi:10.1126/science.abd9149.

5. Rowe BR, Canosa A, Meslem A, Rowe F. Increased airborne transmission of COVID-19 with new variants, implications for health policies. Build Environ. 2022;219:109132. Epub 2022/05/18. doi: 10.1016/j.buildenv.2022.109132. PubMed PMID: 35578697; PubMed Central PMCID: PMCPMC9095081.

6. Zachariah P, Stockwell MS. Measles vaccine: Past, present, and future. The Journal of Clinical Pharmacology. 2016;56(2):133-40. doi: https://doi.org/10.1002/jcph.606.

7. Tellier R. Aerosol transmission of influenza A virus: a review of new studies. Journal of The Royal Society Interface. 2009;6(suppl_6):S783-S90. doi: doi:10.1098/rsif.2009.0302.focus.

8. Cowling BJ, Ip DKM, Fang VJ, Suntarattiwong P, Olsen SJ, Levy J, et al. Aerosol transmission is an important mode of influenza A virus spread. Nature Communications. 2013;4(1):1935. doi: 10.1038/ncomms2922.

9. Tang JW, Eames I, Li Y, Taha YA, Wilson P, Bellingan G, et al. Door-opening motion can potentially lead to a transient breakdown in negative-pressure isolation conditions: the importance of vorticity and buoyancy airflows. J Hosp Infect. 2005;61(4):283-6. Epub 2005/10/29. doi: 10.1016/j.jhin.2005.05.017. PubMed PMID: 16253388; PubMed Central PMCID: PMCPMC7114940.

10. La Rosa G, Fratini M, Della Libera S, Iaconelli M, Muscillo M. Viral infections acquired indoors through airborne, droplet or contact transmission. Ann Ist Super Sanita. 2013;49(2):124-32. Epub 2013/06/19. doi: 10.4415/ann_13_02_03. PubMed PMID: 23771256.

11. Chow EJ, Mermel LA. Hospital-Acquired Respiratory Viral Infections: Incidence, Morbidity, and Mortality in Pediatric and Adult Patients. Open Forum Infect Dis. 2017;4(1):ofx006. Epub 2017/05/10. doi: 10.1093/ofid/ofx006. PubMed PMID: 28480279; PubMed Central PMCID: PMCPMC5414085.

12. Shen F, Yao M. Bioaerosol nexus of air quality, climate system and human health. National Science Open. 2023;2(4):20220050.

MATERIALS AND METHODS:

2. Minor Revision: The methodology to create the nanocoating is well described. It would additionally be good to add the thickness of the PP used.

R: Thanks, this information was certainly missing.

We add the thickness information is a paragraph before:

In a previous study, we reported the effectiveness of a nanometric AgCu film (called SakCu®) deposited on both sides of a 0.3 mm (± 0.03 mm) thick polypropylene [45] fabric, which is usually used as a filtration material in the PPEs, including the N95 masks.

3. Minor Revision: Moreover, it would be good to understand how deep within the PP does the coating deposit. Hence SEM and EDS at multiple thicknesses till no AgCu is detected within the fiber would be useful but is optional.

R: As you can see from the thickness reported above, the PP fabric is very thin, only 300 micros. The electron beam of the SEM penetrates very deep into the material, and because the layer is only a few nanometers thick, so it is not easy to perform the suggested experiment. We will try depositing a slightly thicker coatings and analyzing the cross section, but it will take us sometime to get the information properly. Thanks for the suggestion.

RESULTS

4. Minor Revision: In line 315 describe in which locations and in how many locations was the EDS collected.

R: Thanks, it was not clearly mentioned. We have changed the paragraph as follows:

The AgCu nanolayer's composition and uniformity were assessed using SEM-EDS (Jeol 7600). For this, pieces of 1 cm diameter were cut randomly from the 18 cm wide PP roll. Fig 3 demonstrates the uniform deposition of the film on the PP fibers. The EDS analysis revealed a composition of 39 ± 7 at.% Ag and 61 ± 7 at.% Cu. The slightly higher Cu content compared to our previous report can be attributed to the different target configuration used. The reported values represent the average measurements obtained from different deposition runs (3) in randomly chosen pieces (3 - 4) of the coated PP. The low standard deviation indicates uniform coating and successful repeatability.

5. Minor Revision: The x-axis for figures 5 and 7 incorrectly spell inhibition as ”inibition”.

R: Thanks a lot for the revision, we just missed it. Figures have been corrected.

DISCUSSION

6. Minor Revision: A potential mechanism for less activity of PhiX174 virus should be elaborated based on understanding the cited work (citation 54 and 56).

R: Thanks, there is not a simple explanation. Some of the microbiologist in the group just say that the DNA viruses are usually more resistant, because the DNA nature itself. We read carefully the cited references, but none of them is providing a definite explanation. Nevertheless, we have elaborated the paragraph with more details. 

New text:

Our results demonstrate that AgCu nanolayer was more effective against the ssRNA bacteriophages (PaMx54, PaMx60, and PaMx61) than to the DNA phage (PhiX174) since complete inactivation was achieved in 2-4 hours while 12 hours were required to inactivate the DNA virus. Similarly, in our previous study, the DNA virus Human Papillomavirus (HPV) subtypes types 16 and18 were not sensitive to the AgCu nanofilm [46]. Studies have reported variations in the susceptibility of viruses to disinfectants or virucides based on their characteristics, such as DNA vs. RNA or enveloped vs. non-enveloped viruses. However, the underlying reasons for these differences remain largely unexplored. For instance, Sagripanti et al. [61] investigated the virucidal activity of Cu2+ ions against a range of enveloped and non-enveloped DNA and RNA viruses. Their findings revealed that Cu2+ ions were highly effective against RNA viruses but had limited efficacy against DNA viruses. Nevertheless, a mixture of copper and peroxide could eliminate all five virus types studied. The authors proposed that this could be attributed to the larger toxicity of Cu+ ions resulting from the interaction with the peroxide or due to the generation of reactive oxygen species [61].

More recently, Soliman et al. [62]. reported that Cu ions failed to inactivate the DNA virus Phi X174 within a pH range of 5-8, whereas the RNA virus, MS2, showed significant reduction in infectivity. In contrast, silver (Ag) ions demonstrated effectiveness against both types of viruses, with the degree of efficacy dependent on pH levels. A possible explanation for these results lies in the electrostatic interactions between the metal ions and the amino acids comprising the viral capsid, a phenomenon closely tied to the unique characteristics of each virus [62]. Nevertheless, Cheng et al. compared the inactivation of MS2 and PhiX174 by nanoscale zero-valent iron concluding that both viruses suffer capsid damage but the nucleic acid of MS2 (RNA) was completely degraded in 240 min, while the DNA in PhiX174 was simply more resistant [63]

7. Minor Revision: Line 451-4522 mentions ”The oral anaerobic species exhibited a higher sensitivity to the AgCu nanolayer, which is an interesting finding”. It should be clarified what makes this observation interesting. Example: Is it the first time this has been demonstrated?

R: Thanks for asking this question. The comment was based on our own experience, but looking into the literature, we found that it has been reported as well.

The paragraph was modified to present a brief explanation.

The antibacterial properties of silver (Ag) and copper (Cu) have long been recognized, and their effectiveness as antimicrobial agents continues to be studied. These elements, whether used individually or in combination as nanoparticles or coatings, have demonstrated bactericidal effects and exhibit antimicrobial synergy [66-72]. For instance, previous studies have reported significant inhibition of various bacterial species, including E. coli, S. aureus, A. baumannii, and Bacillus subtilis, when exposed to nanomaterials composed of Ag and Cu [68], [73]. Moreover, several studies have shown that Ag and Cu metals have a more effective bactericidal effect than other metals; such as Al, Zr, and Ti against E. coli [74] and P. aeruginosa [75]. 

Besides, studies have shown that Ag nanoparticles (Ag-NPs) and Cu nanoparticles (Cu-NPs) have demonstrated higher inhibition percentages against oral bacteria compared to nanoparticles derived from bismuth or zirconium [76-79]. This highlights the superior antimicrobial properties of Ag and Cu in the context of oral bacteria. In this study, we observed that oral anaerobic species displayed a high sensitivity to the AgCu nanolayer. This finding is intriguing because previous research has often reported diminished antibacterial activity under anaerobic conditions [80, 81]. The rationale behind this phenomenon lies in the absence of oxygen during anaerobic conditions, which limits the availability of metallic ions (Cu2+ and Ag+), consequently reducing their toxicity towards microorganisms [76, 82] . However, it's worth noting that under anaerobic conditions, metals can still undergo corrosion in aqueous solutions through an alternative mechanism involving the oxidation of the metal into a metal hydroxide.

---

## [Decision Letter · Decision Letter 1]

13 Nov 2023

Antimicrobial activity of Silver-Copper coating against aerosols containing surrogate respiratory viruses and bacteria

PONE-D-23-20447R1

Dear Dr. RODIL,

We’re pleased to inform you that your manuscript has been judged scientifically suitable for publication and will be formally accepted for publication once it meets all outstanding technical requirements.

Kind regards,

Amitava Mukherjee, ME, Ph.D.

Academic Editor

PLOS ONE

Additional Editor Comments (optional):

Reviewers' comments:

Reviewer's Responses to Questions

**Comments to the Author**

1. If the authors have adequately addressed your comments raised in a previous round of review and you feel that this manuscript is now acceptable for publication, you may indicate that here to bypass the “Comments to the Author” section, enter your conflict of interest statement in the “Confidential to Editor” section, and submit your "Accept" recommendation.

Reviewer #1: All comments have been addressed

2. Is the manuscript technically sound, and do the data support the conclusions?

Reviewer #1: Yes

3. Has the statistical analysis been performed appropriately and rigorously? 

Reviewer #1: Yes

4. Have the authors made all data underlying the findings in their manuscript fully available?

Reviewer #1: Yes

5. Is the manuscript presented in an intelligible fashion and written in standard English?

Reviewer #1: Yes

6. Review Comments to the Author

Reviewer #1: (No Response)

7. PLOS authors have the option to publish the peer review history of their article (what does this mean?). If published, this will include your full peer review and any attached files.

Reviewer #1: No

---

## [Editor Report · Acceptance letter]

29 Nov 2023

PONE-D-23-20447R1 

Antimicrobial activity of Silver-Copper coating against aerosols containing surrogate respiratory viruses and bacteria 

Dear Dr. Rodil:

I'm pleased to inform you that your manuscript has been deemed suitable for publication in PLOS ONE. Congratulations! Your manuscript is now with our production department. 

Kind regards, 

on behalf of

Professor Dr. Amitava Mukherjee 

Academic Editor

PLOS ONE